# The Impact of Domiciliary Dental Care and Oral Health Promotion in Nursing Homes of Older Adults: A Systematic Review

**DOI:** 10.3390/ijerph22050683

**Published:** 2025-04-25

**Authors:** Cibelle Cristina Oliveira dos Santos, Izabelle Muller Lessa Miranda, Katherine Thuller, Karoline Reis Silva, Leonardo Santos Antunes, Fernanda Signorelli Calazans, Bruna Lavinas Sayed Picciani

**Affiliations:** Health Institute of Nova Friburgo, Fluminense Federal University, Nova Friburgo 28625650, RJ, Brazil; cibellesantos@unifeso.edu.br (C.C.O.d.S.); izabellelessa@id.uff.br (I.M.L.M.); katherineabr@id.uff.br (K.T.); karolinereis@id.uff.br (K.R.S.); leonardoantunes@id.uff.br (L.S.A.); fernandasignorellicalazans@id.uff.br (F.S.C.)

**Keywords:** older adults, home care, oral health

## Abstract

**Background:** The global increase in the population older than 80 years has led to a paradigm shift centered in the hospital environment, with the inclusion of domiciliary oral health actions improving quality of life. This review evaluates the effects of domiciliary dental care and oral health promotion in nursing homes of older adults. **Methods:** Seven databases were searched without date restrictions from 15 September to 21 November 2024. A manual search was also performed in the reference lists of the included articles. This research included studies evaluating older adults aged ≥80 years, regardless of sex, who received domiciliary dental care associated or not with oral health promotion, evaluating periodontal condition, dental caries, and the dental and denture plaque index. Regarding data collection and analysis, a risk of bias assessment was performed using RoB 2.0 and RoB 1.0, according to the study design. The level of evidence was assessed using the GRADE tool. **Results:** Of the 2415 studies found, 5 met the eligibility criteria. After quality assessment, one randomized clinical trial presented a moderate risk of bias, and one presented a low risk. Also, two non-randomized studies presented a high risk and one a low risk. The certainty of evidence was classified as low for all outcomes assessed. One study demonstrated a reduction in the caries level of participants. Regarding periodontal and gingival conditions, although the occurrence of deep pockets (greater than 3.5 mm) decreased over time, there was no significant difference between the control and intervention groups. The level of dental and denture plaque showed a slight reduction. **Conclusion:** There is limited evidence that domiciliary dental care in nursing homes for older adults can lead to a reduction in caries levels and that oral health programs can reduce dental and denture plaque in evaluations conducted over periods of up to two years. Although the results show a limited magnitude, this does not diminish the importance of promoting domiciliary oral health care.

## 1. Introduction

The current trend of population aging has brought the need for a paradigm shift related to health and social systems. Amid a major demographic transition, the World Health Organization [1] estimates that by 2030, one in six people worldwide will be 60 years or older, with this population increasing from 1 billion in 2020 to 1.4 billion in 2030 and 2.1 billion by 2050. This projection highlights the need for expanded primary and long-term health care services to ensure longevity and a healthy life [2]. In this context, oral health enables essential functions such as eating and speaking, which are linked to self-confidence, well-being, social interaction, and participation in daily activities [2,3]. However, older adults often face compromised oral health [4] due to issues such as edentulism, difficulties with dental prostheses, periodontal diseases, residual roots, and carious lesions [5,6]. This scenario underscores the vulnerability of older adults in accessing dental care, exacerbated by reduced cognitive function, lack of autonomy, depression, and the absence of caregiver guidance on proper oral hygiene [6,7].

In this context, oral health promotion in nursing homes emerges as a viable option. It is a health care service that plays preventive and educational roles in the oral health of older adults [8]. When associated with domiciliary dental care, treatments performed by dentists, this approach ensures equal access to oral care, providing personalized and specialized treatment [9] and promoting better quality of life for functional and healthy older adults [6,10]. Studies suggest that older adults living in nursing homes often have poor oral-health-related quality of life [10,11] and that one in four seniors prefers domiciliary dental consultations due to access barriers such as functional impairments and cognitive decline [9]. This highlights the significant need for domiciliary dental care, which is often overlooked when assessing the demand for general home health care services [9,12]. Neglecting oral health issues disregards their impact not only on overall health but also on the quality of life and daily well-being of this population [13].

Oral conditions such as the need for dentures, tooth loss, and the use of ill-fitting prostheses—with consequent impairments in masticatory function, aesthetics, and speech—affect not only physical health but also psychological and social well-being, directly impacting the quality of life of older individuals [13]. In response to this issue, domiciliary dental care in nursing homes provides older adults with greater comfort and attention by offering preventive, curative, and educational measures. Additionally, it promotes coordination between health care professionals and family members, allowing for the implementation of a treatment plan tailored to older adults’ needs, ultimately aiming to improve their quality of life [8]. However, there are still few studies addressing the emerging need for implementing domiciliary oral health programs in nursing homes within the context of comprehensive older adult care, considering their oral and general health conditions as well as their quality of life. Therefore, this study aims to assess the effectiveness of domiciliary dental care and oral health promotion in nursing homes. The objective is to demonstrate to caregivers, nurses, dentists, patients, and family members the need to implement this service by qualified professionals trained to work in this field.

## 2. Materials and Methods

### 2.1. Protocol and Registration

This review was conducted in accordance with Cochrane guidelines, and the report was described following the PRISMA framework. This study was registered in the International Prospective Register of Systematic Reviews (PROSPERO) under the ID number CRD42024586468 and performed according to PRISMA guidelines (Preferred Reporting Items for Systematic Reviews and Meta-Analyses Protocol) [14] (Appendix A).

### 2.2. Research Question

The following PICOS question was investigated: “In older adults aged 80 and over, receiving domiciliary dental care associated or not to oral health programs in nursing homes, what are the effects on oral health, considering the level of caries, plaque and periodontal disease?”

### 2.3. Eligibility Criteria

The selection followed the criteria of the PICOS acronym, being the following: population (P), older adults aged 80 and over, regardless of sex, living in nursing homes; intervention (I), oral health preventive measures provided by nurses and caregivers, whether associated with dental care provided by dentists or not; comparison (C), individuals who did not receive oral health programs or individuals evaluated before the intervention; outcome (O), level of caries, periodontal condition, and amount of bacterial plaque on teeth and dentures; and study type (S), randomized or controlled clinical trials. The search excluded studies conducted on populations under 80 years of age who did not receive oral health prevention at nursing homes by caregivers or dentists or who did not present data regarding the assessment of the level of caries, dental and denture plaque, or periodontal disease.

### 2.4. Information Sources

The research was conducted on the databases Embase, Latin American Literature in Health Sciences (LILACS), PubMed/MEDLINE, Scopus, and Web of Science. Gray literature was included, searched in Google Scholar and ProQuest, with manual searches of references from the included studies and consultation with experts. There were no restrictions regarding the publication date or language of studies. Comprehensive searches were made from 15 to 21 September 2024.

### 2.5. Search Strategy and Selection Process

The search strategy was created using MeSH terms, keywords, and Boolean operators (“OR” and “AND”) to cover the PICOS acronym (Appendix A). References were exported to the RAYYAN reference manager [15], where duplicate files were removed. Two independent reviewers (C.S. and K.T.) selected the included articles in two phases. First (Phase 1), both reviewers assessed the titles and abstracts according to the eligibility criteria; second (Phase 2), they examined the full texts of the articles included in Phase 1, selecting the articles using the same criteria as in Phase 1 and then verifying all information found. In cases of discrepancies, a third reviewer (B.L.) was involved before making the final decision in both phases. If important data for the review were missing or unclear, an attempt to contact the corresponding author of the study was made to resolve or clarify the issue.

### 2.6. Data Collection Process

Two independent reviewers (C.S. and K.T.) collected the data from the selected articles. Once selected, they verified the recovered information with the third reviewer (B.L.). The collected information included author, study type, year of publication, country, patient characteristics (sample size, sex, and age), clinical characteristics (central tendency measures referring to oral health indices, including caries prevalence, plaque, and periodontal disease), nursing home characteristics, and key conclusions. Any discrepancies were discussed among the three reviewers.

### 2.7. Study Risk of Bias Assessment

Both Risk of Bias 2 (RoB 2) [16] and 1 (RoB 1.0) [17] are methods utilized to assess the risk of bias in randomized clinical trials, with the goal of ensuring the reliability and quality of evidence in systematic reviews. RoB 1 evaluates seven key domains of bias, categorizing each as low, unclear, or high risk based on the available information and the methodology employed in the assessed study. In contrast, RoB 2 offers a more structured and standardized approach, focusing on five key domains that are classified as low risk, some risk, or high risk of bias. This structure facilitates a more objective evaluation. The assessment was conducted by two reviewers, and any discrepancies were resolved by the third reviewer.

### 2.8. Certainty Assessment

Two intervention comparisons (amount of plaque on the teeth and plaque on the dentures) were assessed in this review according to the GRADE tool (Grading of Recommendations Assessment, Development and Evaluation approach) [18]. The narrative GRADE was classified as not serious, serious, or very serious issues in each one of the five domains evaluated: study design, risk of bias, inconsistency, indirectness, and imprecision. The final classification was rated as high, moderate, low, or very low.

## 3. Results

### 3.1. Identification and Selection of Studies

The search in electronic and manual databases retrieved 2858 studies distributed among the following sources: PubMed (402), Web of Science (293), Scopus (422), Cochrane Library (1.702), Embase (35), Livivo (0), Lilacs (0), and gray literature (4). These records were organized in a flowchart illustrated in Figure 1. After removing 439 duplicate records, 2419 studies remained for screening. In the initial analysis, based on titles and abstracts, 2409 articles were excluded because they did not meet the pre-established inclusion criteria. Thus, after this screening, 10 potentially relevant articles were selected for full reading. However, three of them were excluded because they did not answer the review question [10,19,20] and two were cross-sectional studies [21,22]. At completion of the process, five studies were included in the final analysis [23,24,25,26,27]. Additionally, a manual search was performed in the reference lists of the selected articles, but no additional studies were identified.

### 3.2. Assessment of the Methodological Quality of Included Studies

Analysis of the methodological quality of the included studies revealed that the risk of bias was classified as “low” in two studies [23,24], “moderate” in one study [25] and “high” in two studies [26,27], as illustrated in Figure 2 and Figure 3. Concerning random sequence generation, all five studies had a low risk of bias. In two studies the interventions were not described clearly enough [25,27], and another two studies were compromised by missing data [25,26]. One study presented losses throughout the evaluation due to the death of some participants and deviation from intended intervention due to changes in care dependency during the study period [26] The clarity of outcome measurement was considered insufficient in two studies [26,27]. Regarding the inclusion and exclusion criteria, all studies demonstrated a low risk of bias, indicating a robust process of participant selection. 

### 3.3. Data Extraction and Characteristics of Selected Studies

Data extraction from the selected studies is described in Table 1. Two studies were classified as randomized clinical trials, two as controlled clinical trials, and one as an observational cohort trial, involving samples of 269 to 1987 patients of both sexes. Four of the selected studies analyzed the level of dental plaque and the presence of plaque on dentures [23,24,25,27]. Regarding the presence of caries, one study performed this evaluation [24]. The follow-up period varied, with annual assessments performed in some studies [21,22], while others described biannual follow-ups, including regular visits [23] and reassessments after the initial consultation [24]. Participants were assessed based on protocols that included oral health instructions, oral hygiene practices, and personalized interventions [23,24,25,26,27,28]. The articles employed different statistical methods, such as mixed model analysis [23,25,27], *t*-test and chi-square [25,27], a generalized linear model [24,25], a generalized mixed model [27], and the Wilcoxon test [25,26]. Only one study used ANOVA [27], the McNemar test [26], and a generalized linear model [23]. In addition, two studies used logistic regression [24,26]. In total, four studies demonstrated statistically significant results related to the interventions performed, with emphasis on the reduction of dental plaque and the presence of plaque on dental prostheses [24,25,26,27].

### 3.4. Synthesis of Results

A meta-analysis was not feasible in this systematic review due to clinical, methodological, and statistical heterogeneities. The included studies used different protocols of intervention, follow-up periods, measurement methods, and could not be compared by a meta-analysis.

### 3.5. Level of Evidence

According to GRADE, the evaluation of the certainty of evidence is described in Table 2. The certainty of evidence was classified as low. This was mainly due to limitations such as intervention heterogeneity, different assessed methods, and the risk of bias results. Therefore, confidence in the estimate of effects is limited.

## 4. Discussion

Concerning the oral health condition of older adults, an analysis of comorbidities, cognitive impairments, and their consequent impact on quality of life highlights the importance of having trained professionals in the nursing home environment attentive to the individual’s needs [28]. Therefore, it is reasonable to assume that the implementation of domiciliary dental care and oral health programs in nursing homes for older adults would improve oral health indicators. However, the studies evaluated in this systematic review show only low evidence of slight improvements in the oral health of older adults who received domiciliary oral health programs in nursing homes or minimal dental intervention by dentists [23,24,25,26,27].

One study with considerable risk of bias assessed the reduction from 70.5% to 36.5% regarding the caries levels in older adults, who received oral health preventive measures at nursing homes [26]. The domiciliary dental care performed by dentists through restorative procedures plays a fundamental role in reducing the level of caries. The presence of a dentist in the nursing home to provide curative care is essential to ensure greater access to dental care and promote better oral health indices. Regarding the periodontal condition, which involves more extensive interventions such as scaling or even surgeries, it was predictable that merely providing oral health education programs at nursing homes would not be sufficient to promote significant improvements. Although one study with considerable risk of bias [27] showed a reduction in periodontal pockets of approximately 3 mm in depth, this finding does not reflect significant differences between the intervention and control groups after 12 and 24 months, respectively (*p* = 0.62–0.69; 95% CI = −15.6 to 11.8).

Even though there was a slight improvement in oral health, it is essential to also consider the emotional health and well-being of older adults, which typically show low levels of quality of life at the onset of domiciliary dental care [10,11] and which are improved after interventions [29]. In this regard, since the need for treatment and the costs increase with age [19], as well as the degree of dependence [30], oral health preventive measures should be offered at least semiannually and involve all key participants, including caregivers, nurses, family members, and the older adults themselves.

In relation to dental plaque levels, three studies [23,24,25] showed an average reduction in dental plaque indices ranging from 1.4 (32.4) [27] to 1.7 (8.05) [24], a small magnitude associated with a large variability observed in the standard deviation, which makes its clinical significance less representative. The loss of manual dexterity and functional capacity along with the presence of cognitive difficulties and other psychomotor limitations [31] has a significant impact on the ability to perform oral hygiene actions [32], increasing the need for home assistance from a qualified professional [24]. Conversely, professionals without dental training [23,24,25], those who do not practice oral health care themselves, or even those who received oral health information infrequently [27] may not adequately assist the older adults in maintaining oral health. It is undeniable that there are various difficulties faced by nurses and caregivers, including patient resistance to adopting new hygiene habits and varying degrees of dementia, depression, anxiety, and other cognitive and functional limitations [23]. It is also important to consider the work routine of the caregiver, the number of working hours, the number of patients assisted, and pressures faced by family members and managers, among other factors.

Concerning the level of denture plaque, two short-term studies [23,27] demonstrated an improvement in the indices, yet without significant differences between the control and experimental groups. This result is like the evaluation of plaque levels on natural teeth, although there is greater ease in cleaning dentures compared to natural teeth [24]. The use of auxiliary instruments, such as ultrasonic devices for denture cleaning or electric toothbrushes, could help control oral hygiene in older adults; however, these resources require financial investment. Thus, socioeconomic factors can impact oral health conditions, particularly in older adults with fewer financial resources or those residing in institutions that routinely receive little governmental financial support for oral health measures despite the legal obligation to allocate such investment [33].

A deeper understanding of the results of this systematic review should consider not only aspects related to patient limitations but also those of dental professionals. There is a need for specific training to qualify domiciliary dental care for patients with special needs, such as older adults. However, there are few professional courses available, and the demand for this dental specialty remains low despite its growing need. It is essential to encourage dental students, starting from their undergraduate studies, to learn about and engage with this increasingly present reality.

It is also important to consider the need for adequate remuneration for professionals, corresponding to the complexity of care provided, as significant financial investments are required for mobile dental treatment units, consumable materials, and the professional transportation to perform the service [34]. For populations in economically underprivileged regions, this service should be offered by public health systems and promoted by governmental health organizations. Finally, the presence of specialized dental professionals in nursing homes—not only for preventive measures but also for curative actions—working in collaboration with other multidisciplinary professionals at the facility, with a reasonable frequency, can lead to better oral health outcomes for the older adults, improving their quality of life [27].

### Limitations

The greatest difficulties of this systematic review are associated with the lack of studies related to the oral health of older adults receiving domiciliary dental care associated or not with oral health programs in nursing homes. In addition, the literature currently available is quite heterogeneous and has problems related to methodological quality. Thus, the evidence generated by this review is limited. The absence of the control group was a limitation presented by one study in this review [26]. The samples evaluated had varying degrees of dependence, which led to inconsistency and imprecision in the results. Randomization had limitations in some studies and was not sufficient to eliminate confounding factors [23,25]. Thus, the level of evidence generated by the GRADE tool for the evaluations of the plaque levels on both teeth and dentures was considered low due to the considerable risk of bias in the included studies [23,24,25,27]; the inconsistency and imprecision [23,24,25,27] of the results; the different effect estimates used, measured by professionals with different technical backgrounds [23,25,27]; uncontrolled confounding factors, such as systemic medications [27]; and the impact of random error presented in the statistical analysis of studies. However, it is important to highlight that, despite the methodological limitations, the studies indicate an improvement, albeit modest, in the oral health condition of older adults receiving nursing home care, a finding that can be reinforced by new randomized, multicenter clinical studies, where domiciliary dental interventions and evaluations are conducted by dentists at least semiannually, and which control for confounding factors such as the level of dependence, systemic comorbidities, and cognitive difficulties. Among the strengths of our review, we emphasize the transparency of the project, which was conducted in accordance with its registration on the Prospero platform, as well as the methodological rigor applied in its execution and reporting, adhering to the PRISMA guidelines. Additionally, the absence of restrictions in the search for primary studies helps to minimize potential publication bias.

## 5. Conclusions

There is limited evidence that domiciliary dental care in nursing homes for older adults can lead to a reduction in caries levels. Also, oral health programs can reduce dental and denture plaque, with no significant effect on periodontal health, in evaluations conducted over periods of up to two years. Although the results show a limited magnitude, this does not diminish the importance of promoting domiciliary oral health care. Additionally, it is essential to expand the domiciliary dental care, especially for older adults, whose demand has been increasing over the years.

## Figures and Tables

**Figure 1 ijerph-22-00683-f001:**
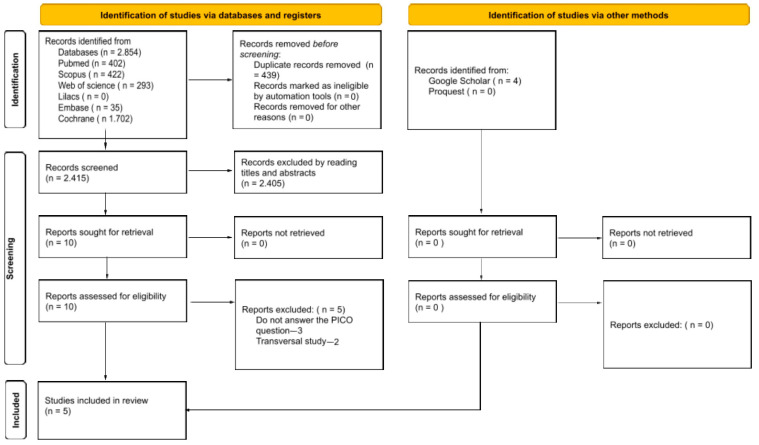
Flow diagram of study identification.

**Figure 2 ijerph-22-00683-f002:**
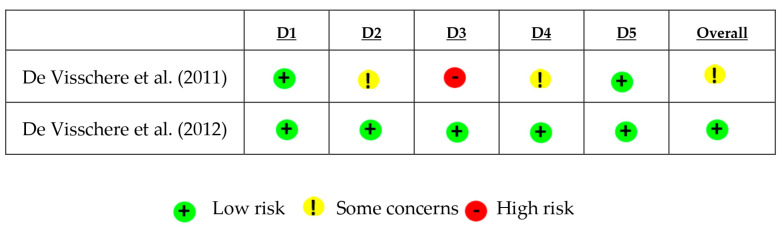
Risk of bias domains among randomized trials, divided into risks of bias: arising from the randomization process (D1); due to deviations from intended interventions (D2); due to missing outcome data; in outcome measurement (D4); and in selection of the reported result (D5) (ROB-2) [23,25].

**Figure 3 ijerph-22-00683-f003:**
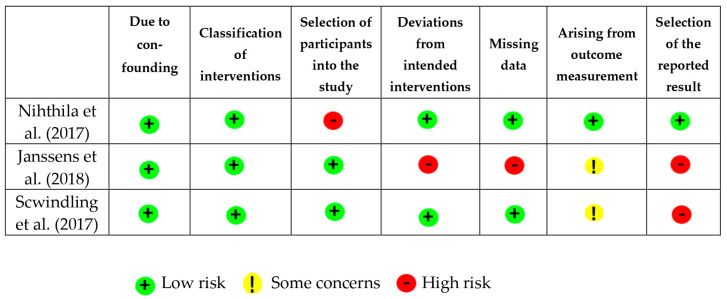
Graph of risk of bias in the reviewer’s judgement about each risk of bias domain across included studies (ROB-I) [24,26,27].

**Table 1 ijerph-22-00683-t001:** Characteristics of included studies.

1. Authorship	2. Materials and Methods		3. Results					4. Conclusions
Author, Year, Study Design, Country	Sample characteristics Sample size, sex (F/M), mean age (standard deviation)	- Characteristics of Nursing Home Services - Intervention- Dental conditions assessed - Statistical analysis	Clinical Dental Characteristics–Oral Health Indices: Dental caries Periodontal condition Gingival condition Dental plaque Plaque on dental prostheses					
De Visschere et al., 2011 [25]. Randomized Clinical Study, Belgium	- 1393 - 75.9%/24.1% - 84.79 years (7.87)	- Implementation of the Flanders-Belgium (2003) oral hygiene protocol in nursing homes, including annual evaluations over 5 years. - Dental plaque levels (Silness and Löe Plaque Index) and denture plaque levels (Augsburger and Elahi Scale) - Multiple regression, *t*-tests, ANOVA, and chi-square test	- NR					There was an initial reduction in plaque levels on prostheses during the first two years and in dental plaque levels after five years of follow-up in patients who received the intervention. However, the plaque levels achieved were statistically and clinically insignificant.Oral hygiene was worse among more dependent residents, and factors such as the presence of toothpaste and caregivers’ ability to assist in oral health habits were associated with cleaner dentures.
			- Dental plaque levels (*n*; mean (SD))					
			Baseline		Control		Intervention	
					46; 2.21(0.81)		- -	
			2-year follow-up		40; (0.80)		13; 1.97 (0.92)	
			5-year follow-up		15; 2.05 (0.87)		28; 1.68 (0.74)	
			- Denture plaque levels (- *n*; - mean (SD))					
			Baseline		Control		Intervention	
					124; 2.06 (0.85)		-	
			2-year follow-up		93; 1.78 (0.78)		32; 1.57 (0.74)	
			5-year follow-up		26; (1.04)		66; 2.05 (0.97)	
De Visschere et al., 2012 [23]. Randomized clinical study, Belgium.	- 373 - 73.2%/26.8% - 84.8 years (8)	- Implementation of the Dutch Oral Healthcare Protocol (2007) in nursing homes, with caregiver training and monitoring in oral hygiene practices during 6 months. - Dental plaque index using the “Silness and Löe” method on a subset called “Ramfjörd teeth”, denture plaque index using the “Augsburger and Elahi” method, and tongue plaque index by Winkel. - Mann–Whitney, Kruskal–Wallis, Wilcoxon, and Spearman tests and regression models.	NR					Considering the low baseline values for different plaque levels, it was concluded that oral hygiene improved with the supervised implementation intervention, although it did not reach the expected improvement. The most satisfactory results were observed for denture plaque, followed by tongue plaque and dental plaque. In addition to the intervention, an increase in dental plaque was observed with a rising dependency scale, while a decline in cognitive function was associated with an increase in dental plaque and a decrease in denture plaque.
			- Plaque levels - Tongue plaque					
			Mean (SD)		Baseline	Follow-up mean (SD)	Adjusted difference	
			Intervention (*n* = 139)		4.25 (4.05)	3.66 (4.19)	−0.07	
			Control (*n* = 139)		4.14 (4.11)	3.66 (4.10)		
			Dental plaque					
			Mean (SD)		Baseline	Follow-up mean (SD)	Adjusted difference	
			Intervention (*n* = 40)		1.60 (0.68)	1.57 (0.79)	−0.15	
			Control (*n* = 57)		1.64 (0.66)	1.77 (0.75)		
			Denture plaque					
			Mean (SD)		Baseline	Follow-up mean (SD)	Adjusted difference	
			Intervention (*n* = 95)		2.19 (0.93)	2.01 (1.00)	−0.32	
			Control (*n* = 97)		2.24 (0.91)	2.37 (1.00)		
			- NR					
Janssens et al., 2018 [26]. Longitudinal cohort study, Belgium	- 381 - 275/106 - 82.4 years (8.9)	The interventions included oral health education by nursing home staff, regular visits, and curative care by dentists using mobile dental equipment, during approximately 22.5 months.	- D3MFT					The oral health program reduced the number of decayed teeth and the number of residents with dental caries, helping a considerable proportion of residents to maintain stable oral health without the need for additional treatments.
			(*n* = 263)		Baseline Mean (median) or number		Follow-up Mean (median) or number	
			Decayed teeth		3.02 (2.00)		1.40 (2.95)	
			Missing teeth		18.90 (19.00		21.86 (23.0)	
		- Presence of dental caries, assessed by the D3MFt index. - Wilcoxon test, McNemar test, and logistic regression.	Filled teeth		1.62 (0.00)		1.89 (1.00)	
			Residents with caries					
			Initially		70.5%			
			End of follow-up		36.5%			
			- NR					
Nihtila et al., 2017 [24]. Controlled clinical study, Finland.	- 269 - 198/71 - 84.55 (5.4)	The intervention was conducted by nurses after receiving instructions from the dental hygienists.	- NR					The intervention had a small positive effect on oral hygiene, reducing plaque on 1.7 teeth. However, the number of teeth with plaque remained high. Multiple approaches based on individual needs are necessary to improve the oral health of vulnerable older adults, including the integration of preventive dental care into the daily care plans conducted by home care nurses.
			- Teeth with plaque					
			Mean (SD)	Baseline		Final	Change	
		- Dental plaque index using the modified “Silness and Löe” method. - Linear and logistic regression models.	Intervention (*n* = 140)	9.5 ± 8.9		7.8 ± 7.2	−1.7	
			Control (*n* = 105)	9.2 ± 7.5		9.4 ± 7.6	0.2	
			- NR					
Schwindling et al., 2017 [27]. Controlled clinical study, Germany.	- 269 - 189/80 - 83.35	The intervention was carried out in nursing homes by caregivers—who received oral health education—and ultrasonic cleaning devices by a dentist over 6 to 12 months.	- NR					- After 6 months of the intervention, patients showed a slight improvement in plaque indices on both teeth and dentures, which was maintained at the 12-month follow-up. - Oral health education programs for caregivers can help maintain the oral health of elderly individuals. - Ultrasonic auxiliary devices can aid in denture hygiene, especially in individuals with significant cognitive impairment.
			- CPITN Grade 3					
			*n*%	6 months *N* = 140		12 months *N* = 94		
				Yes	No	Yes	No	
			Intervention	81 (88.0%)	11 (12.0%)	51 (76.1%)	16(23.9%)	
			Control	40 (83.3%)	8 (16.7%)	23 (85.2%)	4(14.8%)	
			Grade 4					
			*n*%	6 months *N* = 140		12 months *N* = 94		
				Yes	No	Yes	No	
			Intervention	44 (47.8%)	48 (52.2%)	27 (40.3%)	40(59.7%)	
			Control	19 (39.6%)	29 (60.4%)	10 (37.0%)	17(63.0%)	
			- GBI					
			Mean (SD)	6 months– Baseline *N* = 140		12 months– baseline *n* = 94	12 months– 6 months *N* = 94	
		- Periodontal treatment needs (Community Periodontal Index of Treatment Needs—CPITN), gingival bleeding (Gingival Bleeding Index—GBI), plaque accumulation on natural teeth (Plaque Control Record—PCR), and denture hygiene (Denture Hygiene Index—DHI). - *t*-test, chi-square test, and multiple regression models.	Intervention	−6.8 (34.8)		−11.7 (33.9)	2.6 (36.3)	
			Control	−4.0 (31.4)		−4.0 (36.1)	−7.0 (34.1)	
			Group difference	−7.9 (−18.0;2.3)		−6.9 (−21.7;7.9)	2.7 (−14.3;19.7)	
			- PCR					
			Mean (SD)	6 months – baseline *n* = 140		12 months– baseline *n* = 99	12 months– 6 months *N* = 99	
			Intervention	−14.9 (26.3)		−15.5 (27.8)	1.4 (32.4)	
			Control	−0.5 (19.0)		3.5 (18.5)	−0.4 (16.1)	
			Group difference	−14.4 (−21.8; −6.9)		−16.2 (−27.7; −4.7)	1.0 (−11.6;13.)	
			- DHI					
			Mean (SD)	6 months– baseline *n* = 165		12 months– baseline *n* = 114	12 months– 6 months *N* = 114	
			Intervention	−26.0 (28.3)		−27.4 (29.3)	−1.5 (26.4)	
			Control	−6.0 (18.7)		−8.3 (24.7)	−2.6 (22.0)	
			Group diference	−15.0 (−23.6; −6.5)		−13.3 (−24.9; −1.8)	−2.4 (−13.9; 9.1)	

CPITN = Community Periodontal Index of Treatment Needs; DHI = Denture Hygiene Index; D3MFT = Decayed, Missing, and Filled Teeth at the D3 threshold; F = Female; GBI = Gingival Bleeding Index; M = Male; N = sample; NR = not reported; PCR = Plaque Control Record; SD = standard deviation.

**Table 2 ijerph-22-00683-t002:** Quality assessment of included studies.

Assessment of the Certainty of Evidence							Impact	Certainty
№ of Studies	Study Design	Risk of Bias	Inconsistency	Indirect Evidence	Imprecision	Other Considerations		
Change in the amount of dental plaque								
4	Observational study	Serious ^a^	Serious ^b^	Not serious	Serious ^c^	All potential confounding factors reduced the demonstrated effect	Among the four studies that assessed the amount of dental plaque, three evaluated the individuals 6 months after the intervention and reported statistical differences in dental plaque levels [23,24,27], although with little clinical significance and no differences between groups. The study by De Visschere et al., 2011 [25], shows a positive result in reducing plaque levels after 5 years of follow-up.	⨁⨁◯◯Low
Change in the amount of plaque on dental prostheses								
3	Observational study	Very serious ^a^	Serious ^d^	Not serious	Not serious	All potential confounding factors reduced the demonstrated effect	The studies present different evaluation periods and report inconsistent results. De Visschere et al., 2012 and 2011 [23,25], report reductions in denture plaque levels at 6 months, while Schwindling et al., 2017 [27], reports that the first study states that the denture plaque index decreases 2 years after the intervention, and the second study reports no change in the amount of denture plaque 1 year after the intervention.	⨁⨁◯◯Low

Explanations: ^a^. The study by De Visschere et al., 2011 [25], presents limitations in the allocation of individuals to groups. The study by Schwindling presents confounding factors, such as systemic medications, which were not controlled and may have impacted the results. ^b^. The studies present different estimates of effect, measured by different types of professionals. ^c^. There is a considerable impact of random error on the certainty of the evidence generated based on the assessment of studies. ^d^. The studies by De Visschere et al., 2011, and Schwindling et al., 2017 [25,27], have different follow-ups. While De Visschere et al., 2011 [25], report an improvement in the denture plaque index after 2 years of intervention, the study by Schwindling et al., 2017 [27] reports a decrease in the index 6 months after intervention onset, with no change after 1 year following the intervention.

## Data Availability

No new data were created or analyzed in this study. Data sharing is not applicable to this article.

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
