# Peer review of "The Impact of Domiciliary Dental Care and Oral Health Promotion in Nursing Homes of Older Adults: A Systematic Review"

_ijerph, 2025, doi:10.3390/ijerph22050683_

Round 1
Reviewer 1 Report
Comments and Suggestions for Authors
This is a systematic review on the effect of 'home care' on the oral health of frail older adults. The methodology used in the review seems sound, but I have a major issue with the research question. To me, the 'I' of the PICO is not defined well, so you end up with included studies with completely different sort of interventions. What was your aim? Please clarify this, the rationale and the very vague, difficult to interpret terminology.
Home care, home dental care, oral health guidance: to me it is not clear what is now the intervention from the abstract and title. All these terms can be interpreted completely differently.
Introduction: line 45-53 says more or less the same as line 54-61, maybe collate the information into one paragraph.
From your introduction, I’m still not sure what you mean by ‘home dental care’. Is it domiciliary dental care (professional dental care at the patient’s home) or is it oral care provided by home care nurses? I would suggest to make this a lot clearer.
‘improving the systemic health of elderly’ -> was this also an aim of the systematic review? Seems quite ambitious, as we know there aren’t many studies on domiciliary dental care. Maybe the aim of ‘improving oral health in frail older adults’ is ambitious enough? And relevant on its own without direct claims to systemic health.
Terminology: elderly = older adults, older people, older persons.
Research question: home care? What is home care? To me, its home care ‘nursing’.
The interventions conducted by De Visschere were solely of a preventive nature: this is oral health promotion. Komulainen and Janssens’ interventions comprise professional dental care (true domiciliary dental care). Schwindling is somewhere in between. I fear this is also due to the major issue in defining the term ‘home care’.
Can you explain table 3: The study by Janssens et al., 2018, presents divergences in the execution planned for the intervention and losses throughout the study. ?
In your discussion: home care is now: ‘health promotion or minimal dental interventions’. Line 243. ? You cannot compare these DDC interventions to oral health promotion interventions?
‘The Jansen study reports a decrease of 1 to 2 decayed teeth (pË‚0.001), on average, 247
after 2 years of applying oral health guidance measures provided by dentists. However, 248
among the study's weaknesses are the absence of a control group, the increase in the num- 249
ber of lost teeth, and the involvement of dentists who performed minor restorative proce- 250
dures—uncontrolled confounding factors that may have influenced the results.’
The actual intervention here was the combination of curative and preventive care. Thus ‘restorations provided by the dentists’ are not confounding factors, they are the actual intervention.
The paragraph where inferences are made on systematic health I’m not sure of, you found nothing on systemic health in your review, why add this to the discussion?
Terminoly: community homes (??), institutionalised elderly, nursing homes: use one term for long-term care settings, and one for community-dwelling older adults if necessary.
General comment on the discussion: results are repeated a lot.
I’m missing the limitations of your own review, not only the limitations of the included studies.
Conclusion: based on what data do you conclude that there is a need for greater training of dental professionals? (I agree but I not finding this in your review). I would suggest to adapt the conclusion to the data found in your study.
Comments on the Quality of English LanguageOverall the English is wordy, with here and there language errors. I would suggest a native speaker to take a critical look.
Reviewer 2 Report
Comments and Suggestions for Authors
I congratulate the authors for their successful systematic review and the manuscript they have produced as a result. Upon reviewing the manuscript, it is evident that they have conducted a highly comprehensive and detailed literature review. Furthermore, as the authors have pointed out, access to healthcare in the community, rather than within hospital settings, is becoming increasingly important for the elderly, as it has a significant impact on their quality of life.
I have a few suggestions for this professionally written manuscript:
- Please use the term "systematic review" in the title.
- Indicate the specific date range for the databases you searched in the Abstract section.
- Include your PROSPERO registration number in the Abstract section.
- The Introduction section is comprehensive and sufficient. I would like to commend the authors in this regard.
- In the Methods section, provide a detailed description of the publication date range of the studies you researched.
- Please explain the linguistic criteria for including the articles you evaluated. It is not expected for authors to be fluent in all languages.
- The Results section, along with the tables and figures, has been prepared in a detailed and professional manner. I commend the authors for this.
- Along with the limitations of your study, the strengths should also be discussed.
- A conclusions section that is accessible to a general reader should be written. It would be more appropriate to make a clear and sociological inference in this section.
Once again, I congratulate the authors and look forward to their revisions.
Round 2
Reviewer 1 Report
Comments and Suggestions for Authors
Thank you for the revisions.
The terminology used ('living in nursing home care') is not what is commonly used in literature and I would suggest adapting it. For an American public 'nursing home', British 'Care home', furthermore 'long-term care' or 'residential aged care facility' is commonly used. I fear you have misinterpreted my question on what is 'home care'. This was a question, where I sincerely was confused what you meant with 'home care'. The confusion persists.
Same with 'oral health guidance': you specifically mean 'oral health promotion interventions'? Specifically on oral health prevention and daily oral care? Sorry but this is still not clear to me. How are you going to see the effect of the oral health promotion apart from the professional dental work done? Seems to me the biggest effect on caries would be the restorative work, not the daily oral care in a relatively short period of time?
Domiciliary dental care is the most commonly used term in literature for 'professional dental care in a (nursing or care) home of the patient'.
Maybe it's interesting to access a native speaker who has some expertise in this area? Looks like a lot of information concerning terminology is lost in translation, and you misunderstood some of my suggestions and questions. An editing service will certainly improve the overall language (and it has), but the terminology should be chosen by the authors.
Comments on the Quality of English LanguageMaybe it's interesting to access a native speaker who has some expertise in this area? Looks like a lot of information concerning terminology is lost in translation, and you misunderstood some of my suggestions and questions. An editing service will certainly improve the overall language, but the terminology should be chosen by the authors.
Author Response
Thank you for your suggestion. please see the attachment
